# Siderophore-Based Molecular Imaging of Fungal and Bacterial Infections—Current Status and Future Perspectives

**DOI:** 10.3390/jof6020073

**Published:** 2020-05-29

**Authors:** Milos Petrik, Joachim Pfister, Matthias Misslinger, Clemens Decristoforo, Hubertus Haas

**Affiliations:** 1Institute of Molecular and Translational Medicine, Faculty of Medicine and Dentistry, Palacky University, 77900 Olomouc, Czech Republic; milos.petrik@upol.cz; 2Department of Nuclear Medicine, Medical University Innsbruck, 6020 Innsbruck, Austria; joachim.pfister@i-med.ac.at; 3Institute of Molecular Biology, Medical University Innsbruck, 6020 Innsbruck, Austria; matthias.misslinger@i-med.ac.at

**Keywords:** imaging, infection, siderophore, positron emission tomography, fluorescence, fungal, bacterial

## Abstract

Invasive fungal infections such as aspergillosis are life-threatening diseases mainly affecting immuno-compromised patients. The diagnosis of fungal infections is difficult, lacking specificity and sensitivity. This review covers findings on the preclinical use of siderophores for the molecular imaging of infections. Siderophores are low molecular mass chelators produced by bacteria and fungi to scavenge the essential metal iron. Replacing iron in siderophores by radionuclides such as gallium-68 allowed the targeted imaging of infection by positron emission tomography (PET). The proof of principle was the imaging of pulmonary *Aspergillus fumigatus* infection using [^68^Ga]Ga-triacetylfusarinine C. Recently, this approach was expanded to imaging of bacterial infections, i.e., with *Pseudomonas aeruginosa.* Moreover, the conjugation of siderophores and fluorescent dyes enabled the generation of hybrid imaging compounds, allowing the combination of PET and optical imaging. Nevertheless, the high potential of these imaging probes still awaits translation into clinics.

## 1. Introduction

Iron is an indispensable metal for virtually all organisms. During infection, pathogens typically encounter iron limitation as the host iron is tightly sequestered by host proteins such as hemoglobin, transferrin, lactoferrin, and ferritin and iron–sulfur cluster proteins [1]. Consequently, pathogens evolved strategies to capture iron from the host sources. *Aspergillus fumigatus* employs two high-affinity iron acquisition mechanisms: reductive iron assimilation and siderophore-mediated iron acquisition [2]. Both systems are induced by iron starvation in vitro as well as in vivo in a murine model of pulmonary aspergillosis [3,4]. The genetic inactivation of siderophore biosynthesis attenuated the virulence of *A. fumigatus* in murine infection models, demonstrating that the siderophore system represents a virulence determinant [2]. Siderophores are low-molecular mass, ferric iron (Fe^3+^)-specific chelators that are secreted by most bacteria and fungi; however, there is considerable species-specific diversity in siderophore structures [5]. *A. fumigatus* secretes two cyclic hydroxamate-type siderophores (Figure 1A), fusarinine C (FsC) and its tri-acetylated derivative triacetylfusarinine C (TAFC). Upon chelating iron, the siderophore-iron complex is taken up by specific transporter, which belong to the Siderophore Iron Transporter (SIT) subfamily of the major facilitator protein superfamily [6]. The presence of SITs is confined to the fungal kingdom and even species that lack the production of siderophores, such as *Saccharomyces cerevisiae*, *Candida albicans*, or *Cryptococcus neoformans*, possess SITs for the uptake of siderophores produced by other species [7]. *A. fumigatus* possesses five potential SITs, of which MirB (major facilitator iron regulated B) was identified as the transporter for TAFC, while the FsC transporter remains to be identified [8]. Within the cell, TAFC and FsC are hydrolyzed by specific esterases [9,10,11]; the released iron is transferred to the metabolism or can be stored either in the intracellular siderophore ferricrocin or within the vacuole [12]. A scheme of TAFC-mediated iron uptake is shown in Figure 2.

Of particular importance, siderophore usage is confined to fungal and bacterial kingdoms. Moreover, bacteria and mammals lack SIT-type transporters [6]. These differences enable fungal-specific in vivo targeting strategies exploiting the siderophore system.

## 2. Proof of Principle: [^68^Ga]Ga-TAFC for Imaging Fungal Infections

New and improved diagnostic methods for invasive fungal infections are needed [13] as current diagnostic approaches, including laboratory tests and computed tomography (CT), have major limitations, both in terms of sensitivity and specificity. For example, in CT, potentially any radiological sign can be accompanied with a fungal invasion. Empirical therapies (i.e., treating patients without diagnosis only based on risk assessment) have been established as the standard of care in many centers around the world, despite the known immediate and long-term consequences related to costs, the development of drug resistance, or toxicity, and without final proof on overall survival.

Imaging modalities providing localization of the infection site with high specificity and sensitivity are challenging to develop. Imaging probes specifically involved in pathophysiological mechanisms of the invading pathogen could be key in this research area. In particular, radiolabeled probes can provide the required sensitivity, and especially positron emission tomography (PET) can also provide appropriate image resolution. PET has evolved as a major clinical imaging technique, particularly in oncology [14], using predominantly 2-[^18^F]-fluorodeoxyglucose, which enables the imaging of glucose consumption. A radiotracer for the specific imaging of fungal infections should ideally fulfill similar criteria: (1) specific involvement in the pathogen’s metabolism with an active uptake mechanism, (2) a trapping process combined with favorable pharmacokinetics, including rapid elimination from healthy tissue, and (3) the pathogen recognizing the radiotracer as an apparent molecule of interest, boosting its active uptake and accumulation from the extracellular matrix.

SITs represent a promising target for molecular imaging approaches in fungal infections for the following reasons: (1) they are highly upregulated during infection; (2) they are not present in human cells, and therefore, their specific substrates do not interact with the human cellular systems; (3) the energy-dependent active uptake leads to the accumulation of a (radio)labeled substrate in the target; (4) the low molecular mass of siderophores (e.g., the Mr of desferri-TAFC is 853 Da) and their high hydrophilicity ensure rapid diffusion from the circulation into infected tissues, but also rapid clearance from non-target tissue and elimination via renal excretion; and (5) the radiolabeling of siderophores can be achieved easily by replacing Fe^3+^ in the siderophore by an iron-mimicking radionuclide. There is no isotope of iron with suitable properties for nuclear imaging in terms of half-life and photon emission. However, Ga^3+^ is an isosteric diamagnetic substitute for Fe^3+^ and has been used extensively to characterize siderophore complexes (e.g., [15]). In the past decade, interest in the isotope gallium-68, a positron emitter, has increased tremendously with the establishment of PET in clinics [16]. Gallium-68 can be obtained from a ^68^Ge/^68^Ga generator, and with a half-life of 68 min, it exhibits a very low radiation burden for the patient.

Proof of concept studies demonstrated that a variety of desferri-siderophores can be radiolabeled with gallium-68 using only micrograms of the siderophore with high chemical stability [17,18]. Siderophore uptake by *Aspergillus fumigatus* was upregulated under iron starvation conditions and could be blocked with an excess of siderophore or NaN_3_, indicating specific and energy-dependent uptake. Only [^68^Ga]Ga-TAFC and [^68^Ga]Ga-desferrioxamine E (DFO-E), a bacterial xenosiderophore, displayed a good combination of fungal uptake in culture, high metabolic stability, and suitable pharmacokinetics for imaging [19]. The high contrast imaging of *A. fumigatus* pulmonary infection in a rat model was achieved using µPET/CT technology, exhibiting the pronounced accumulation of [^68^Ga]Ga-TAFC in infected areas (Figure 1B,C). Significant accumulation of [^68^Ga]Ga-TAFC was found neither in sterile inflammations nor in tumor cells [20]. Notably, the utilization of siderophores displays a certain species specificity. In vitro studies revealed the uptake of [^68^Ga]Ga-TAFC by *A. fumigatus*, *Rhizopus oryzae*, and *Fusarium solani*, but no significant uptake by *Aspergillus terreus, Aspergillus flavus*, *C. albicans* or the bacterial species *Pseudomonas aeruginosa*, *Klebsiella pneumoniae*, *Staphylococcus aureus*, and *Mycobacterium smegmatis* [20]. In comparison, [^68^Ga]Ga-DFO-E displayed the highest uptake by *A. fumigatus* and considerable uptake by *A. terreus*, *A. flavus*, *R. oryzae*, *F. solani*, as well as the bacterial species *S. aureus* [20]. Taken together, in contrast to DFO-E, TAFC appears to be fungal-specific. From these studies, we concluded that ^68^Ga-labeled siderophores, in particular [^68^Ga]Ga-TAFC, have a high potential as radiotracers to image invasive *A. fumigatus* infections in patients.

## 3. Siderophore Modifications

Stimulated by the successful application of [^68^Ga]Ga-TAFC for molecular imaging with PET/CT, chemical modification of these molecules were designed for new potential applications (Figure 2). For example, by attaching fluorescent dyes, a hybrid imaging compound can be created allowing PET/CT with gallium-68 combined with optical imaging (Figure 3). Such a compound might allow image-guided procedures using the optical signal, for instance surgical probing or bronchoscopy. Initial attempts were made to chemically modify TAFC with neutral and positive/negative charged functional groups to investigate its recognition by the MirB transporter in *A. fumigatus* [21]. Chemical modifications were possible even without impairing fungal uptake, while best results were observed for the diacetylated form of FsC (DAFC), whereupon the free amine was modified with functional groups. Based on these findings, fluorescent dyes were coupled to DAFC to test the concept of hybrid imaging [22]. Figure 3A shows a PET/CT scan of a rat with pulmonary *A. fumigatus* infection [23], visualized by the radioactive tracer [^68^Ga]Ga-DAFC-Cy5. Infection can be clearly depicted from the PET signal in the lung of the animal. In addition, optical imaging of the whole lung after dissection reveals a clear corresponding fluorescence signal in the infected lung (Figure 3B). When labeled with iron, these fluorescent siderophores could be also used for fluorescent microscopy, as shown in Figure 3C with [Fe]DAFC-Cy5. The compound was taken up by hyphae and accumulated internally in longitudinal cellular compartments. However, target recognition and pharmacokinetics were highly dependent on the fluorophore coupled to DAFC and should be therefore considered with care. Nevertheless, this concept could be expanded to couple antifungal compounds and use these to combine diagnosis (labeled with gallium-68) with therapy with the same molecule, attempting a so-called “theranostic” approach [24]. These conjugates would allow a targeted antifungal therapy of *A. fumigatus*-infected patients utilizing a Trojan horse concept [25], while similar concepts have been attempted to develop novel bacterial antibiotics [26,27]. Noteworthy, a FsC-Cy5.5 conjugate was also applied for the imaging of skin infections caused by *Trichophyton rubrum* and *C. albicans* [28].

The siderophores used in these studies share one common property: they are all fermentation products naturally produced by fungi or bacteria. Siderophores from synthetic origin would simplify the translation into clinical applications. This has prompted researchers to develop artificial siderophores, thereby identifying minimal essential features and incorporating them into the simplest possible molecular structure, e.g., [29]. A wide variety of biomimetic analogs has been synthesized and evaluated, e.g., ferrichrome analogs [30]. Rationally engineered synthetic siderophores, recognizable and utilizable by the same uptake system as the natural siderophores, provide a unique strategy to develop novel diagnostic and therapeutic tools in infectious diseases.

## 4. Other Siderophores for Molecular Imaging of Microbial Infections with PET

The use of iron as well as other metals as cofactors in basic metabolic pathways is essential to both pathogenic microorganisms, including bacteria and their hosts. Bacterial pathogens employ a combination of up to five primary mechanisms to satisfy the requirement for nutrient iron including the siderophore-mediated iron uptake, heme acquisition systems, transferrin or lactoferrin receptors, and ferrous iron uptake. Overall, most pathogenic bacteria possess several strategies for exploiting host iron sources with certain species specificity [31,32,33,34,35,36]. An important player in this context is siderophore-mediated iron uptake.

We have shown that the radiolabeling of siderophores utilized by bacteria including pyoverdine (PVD), desferrioxamines (DFOs), and ferrichromes with gallium-68 or even zirconium-89 is achievable with high radiochemical purity and stability [18,37,38]. Most of the radiolabeled siderophores revealed favorable in vitro characteristics and rapid pharmacokinetics with renal excretion in healthy animals. Detailed preclinical testing including in vitro assays in various microbial cultures as well as the imaging of infected animals was carried out primarily with [^68^Ga]Ga-PVD. PET/CT imaging of *P. aeruginosa* infection was performed with [^68^Ga]Ga-PVD in two independent animal models: a murine myositis and rat pulmonary model (Figure 4).

The investigations of [^68^Ga]Ga-PVD for *P. aeruginosa* infection imaging, summarized in Petrik et al. [38], demonstrated that [^68^Ga]Ga-PVD can be used for the detection of *P. aeruginosa* infection with high specificity and sensitivity, as even infections with very low burden of bacteria were visible by PET/CT. [^68^Ga]Ga-PVD also displayed much better pharmacokinetics than radiopharmaceuticals clinically used for infection imaging [38,39]. PVD is exclusively produced by *Pseudomonas* spp., and uptake appears to be limited to this bacterial genus. In agreement, in vitro studies indicated a lack of PVD uptake by the bacterial species *Acinetobacter baumannii*, *Burkholderia multivorans*, *Burkholderia cenocepacia*, * Escherichia coli*, *K. pneumoniae*, *Listeria monocytogenes*, *S. aureus*, *Stenotrophomonas maltophilia*, *Streptococcus agalactiae*, and *Yersinia enterocolitica* as well as by the yeast *C. albicans* [38]. Moreover, *A. fumigatus* was shown to be unable to utilize PVD [40]. The additional use of radiolabeled siderophores for infection targeting was notified by Ioppolo et al. [41]. They exploited desferrioxamine-B (DFO-B) for ^67^Ga-labeling and included modifications to adjust pharmacokinetics for imaging *S. aureus* infections. They could show active uptake in bacterial cultures in vitro and selective uptake in infections in vivo but did not provide any imaging data. These data demonstrate that the concept of radiolabeled siderophores for specific infection imaging is transferable to different microbial pathogens.

## 5. Siderophore Diagnostics from Urine

On one hand, the rapid elimination of [^68^Ga]Ga-TAFC via the renal system is a certain limitation in depicting infections in the kidney or bladder region by means of PET. On the other hand, these studies stimulated investigations to use TAFC as a urine biomarker for the non-invasive diagnosis of invasive *A. fumigatus* infections. Promising results were reported both in preclinical studies [23] as well as in clinical samples [42]. However, further studies are needed to validate this method before it can be established in clinical routine. Furthermore, the mass spectrometric methodology used is not typically found in routine diagnostic laboratories. Nevertheless, the detection of TAFC in aspergillosis patients proves the activation of the siderophore system also in human infections.

## 6. Conclusions and Outlook

For many years, siderophores have been applied or developed for a variety of medical applications. Molecular imaging applications for invasive fungal and other infectious diseases have been one focus of research with highly promising results in particular to target infections with *A. fumigatus* and *P. aeruginosa*. Based on this concept, new approaches by e.g., modifying siderophores have been developed, and first proof-of-concept studies have been reported revealing the promising prospects of these applications (Figure 2).

## Figures and Tables

**Figure 1 jof-06-00073-f001:**
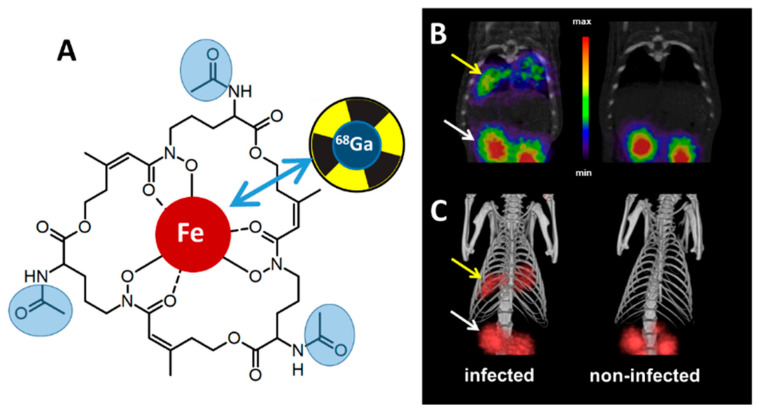
(**A**) Triacetylfusarinine C (TAFC) is shown in the ferri-form; the three acetyl groups lacking in FsC, which can be substituted by other residues for functional modifications, are shown in light blue. For TAFC-based nuclear imaging, the iron (shaded in red) is replaced by gallium-68 (blue double arrow). (**B**) µPET/CT coronal slices and (**C**) volume rendered 3D images of *A. fumigatus* in a rat infection model (thoracic part) after the intravenous injection of [^68^Ga]Ga-TAFC showing clear accumulation (yellow arrows) in infected lung tissue (**left**). Accumulation in kidneys (white arrows) is caused by renal excretion of [^68^Ga]Ga-TAFC; right: control, non-infected rat. CT: computed tomography, PET: positron emission tomography.

**Figure 2 jof-06-00073-f002:**
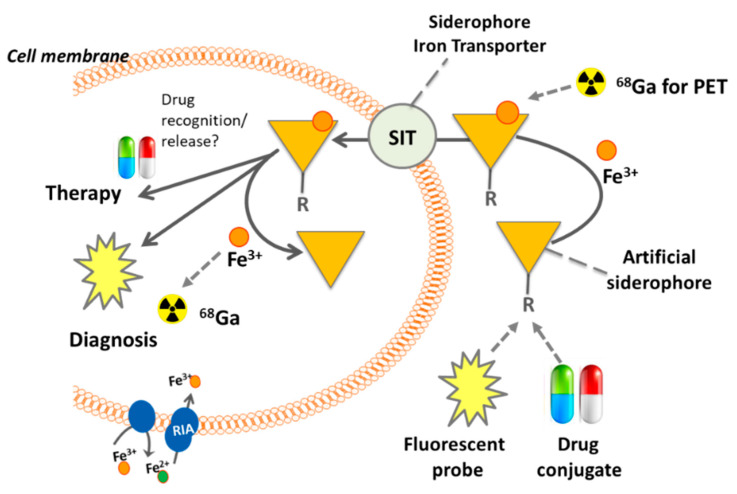
Concept of siderophores for medical translation including the use of natural siderophores, artificial siderophores, modified siderophores attached to fluorescent probes or drugs, and radiolabeled with gallium-68 to allow molecular imaging and/or therapeutic applications. The uptake by siderophores mediates intracellular uptake potentially enabling diagnosis via radioactive or fluorescent signaling or therapy by the introduction of antifungal drugs, thereby following a Trojan horse approach. The dotted line arrows mark replacement of iron by gallium-68 (for PET) or conjugation of fluorescent probes or drug moieties to siderophores (artificial siderophores). The solid line arrows mark the metabolic route and translational goals of siderophores.

**Figure 3 jof-06-00073-f003:**
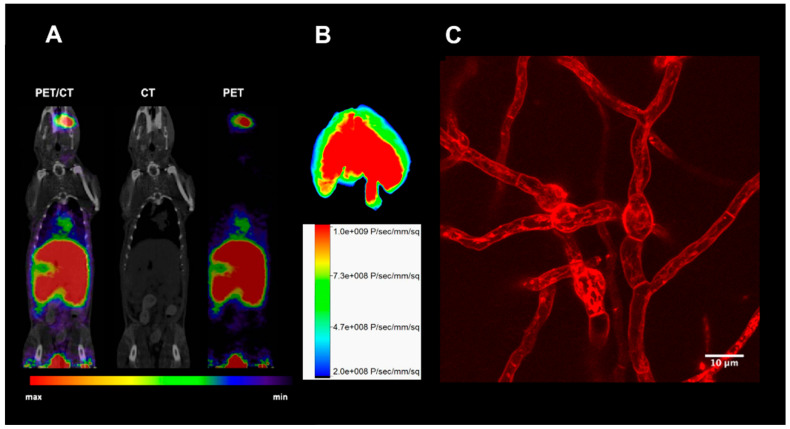
µPET/CT image of [^68^Ga]Ga-DAFC-Cy5 of an immunocompromised Lewis rat infected with *A. fumigatus.* (**A**) Signal in the lung region reflects uptake of radiolabeled siderophore. (**B**) Optical image of the dissected lung; a fluorescence signal can be clearly seen uniformly distributed. (**C**) Fluorescence microscopy image of [Fe]DAFC-Cy5 in *A. fumigatus* hyphae.

**Figure 4 jof-06-00073-f004:**
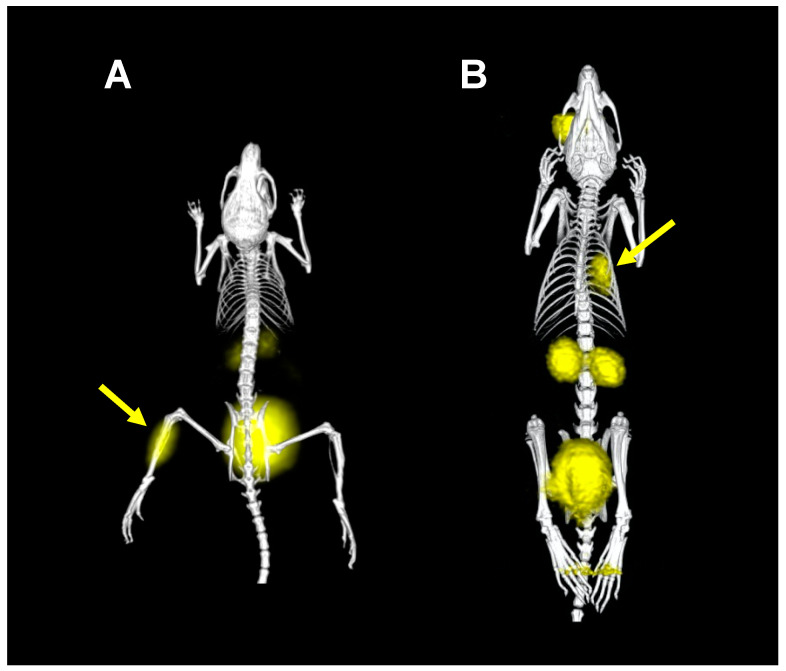
3D volume rendered PET/CT images of [^68^Ga]Ga-PVD in murine myositis (**A**) and rat pulmonary (**B**) *P. aeruginosa* infection models. Yellow areas show accumulation of [^68^Ga]Ga-PVD in infection sites in muscle (**A**) or lung (**B**), which are marked by yellow arrows, as well as in the retro-orbital injection site (**B**) and excretory organs (kidneys and urinary bladder) due to renal excretion of [^68^Ga]Ga-PVD.

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
