# Peer review of "Siderophore-Based Molecular Imaging of Fungal and Bacterial Infections—Current Status and Future Perspectives"

_jof, 2020, doi:10.3390/jof6020073_

Round 1

Reviewer 1 Report

Authors in the manuscript “Siderophore-based molecular imaging of fungal and bacterial infections – current status and future perspectives” presented the review on findings on the preclinical use of siderophores for molecular imaging of infections. Authors presented very interesting and important method for imaging fungal infections, which could provide crucial information on location of infection. Potentially this method could be used for monitoring therapeutic response as well. Even though, the method needs more validation and translational studies, it is very important tool with potential applications.

I have couple specific questions, which I believe will improve manuscript for readership. Please see below:

  1. Line 47: Please spell out MirB, I believe it is first time used on this line. Please make sure, that abbreviations first time used in the manuscript are spelled out.
  2. Line 188-191: Since most elimination is caused by renal excretion of [68Ga]Ga-TAFC, and it is depicted on PET/CT scanning, could it be that renal infection or uninfected animals will show the same accumulation (Fig4)?

Author Response

Dear Editor!

We appreciate the suggestions given by the reviewers to improve our manuscript. We addressed all raised issues and amended the manuscript accordingly. Our specific responses are given below. Thus, we hope that the manuscript now meets the criteria for publication in JOF.

Thank you for your efforts!

Sincerely yours,

Hubertus Haas

Point-by-point response______________________________________________

Reviewer 1

  1. Line 47: Please spell out MirB, I believe it is first time used on this line. Please make sure, that abbreviations first time used in the manuscript are spelled out.

We included the ”historical” abbreviation for MirB and checked all abbreviations.

  1. Line 188-191: Since most elimination is caused by renal excretion of [68Ga]Ga-TAFC, and it is depicted on PET/CT scanning, could it be that renal infection or uninfected animals will show the same accumulation (Fig4)?

Elimination of 68Ga-TAFC and other siderophores predominantely via the renal route is on the one hand favourable as non-specifically bound radiotracer will be quickly eliminated.On the other hand the radioactivity in the renal system will (most likely) overlay potential infectious sites in the kidney or bladder. This is certainly a limitation of this method. This is now mentioned (lines 201-201).

Reviewer 2 Report

In this review article, “Siderophore-based molecular imaging of fungal and bacterial infections – current status and future perspectives”, Petrik et al summarize approaches that utilize siderophores to allow identification of fungal (or bacterial) infection in live animal models. This review is useful and relatively clear. I just have a few comments:

-The authors discuss use of these methods to identify both fungal and some bacterial (ex. P. aeruginosa) infections. The article would benefit from more discussion on how these similar methods can (or might be able to in the future) distinguish between different infections—i.e. if siderophores can be taken up by multiple different pathogens, how could they actually be used as a diagnostic tool? How specific can the targeting of the siderophores be?

- Fig 2 is not actually discussed or cited in the text.

- The article would benefit from a glossary/key of all of the abbreviations used.

Author Response

Dear Editor!

We appreciate the suggestions given by the reviewers to improve our manuscript. We addressed all raised issues and amended the manuscript accordingly. Our specific responses are given below. Thus, we hope that the manuscript now meets the criteria for publication in JOF.

Thank you for your efforts!

Sincerely yours,

Hubertus Haas

Point-by-point response______________________________________________

Reviewer 2

- The authors discuss use of these methods to identify both fungal and some bacterial (ex. P. aeruginosa) infections. The article would benefit from more discussion on how these similar methods can (or might be able to in the future) distinguish between different infections—i.e. if siderophores can be taken up by multiple different pathogens, how could they actually be used as a diagnostic tool? How specific can the targeting of the siderophores be?

Indeed, siderophore uptake displays a certain species specificity. This is now addressed (lines 116-122 and lines 188-193)

- Fig 2 is not actually discussed or cited in the text.

Fig. 2 was referenced in line 51 and is now additionally referenced in lines 127 and 215.

- The article would benefit from a glossary/key of all of the abbreviations used.

Thank you for this suggestion, we included a list of abbreviations at the end of the article.